# Severe hyperbilirubinemia is associated with higher risk of contrast-related acute kidney injury following contrast-enhanced computed tomography

**Yu-Hsien Wu**[1], **Chun-Yi Wu**[2], **Ching-Yao Cheng**[3,4], **Shang-Feng Tsai** [2,5,6]*

**1** School of Medicine, China Medical University, Taichung, Taiwan, **2** Division of Nephrology, Department of Internal Medicine, Taichung Veterans General Hospital, Taichung, Taiwan, **3** Department of Pharmacy, Taichung Veterans General Hospital, Taichung, Taiwan, **4** Department of Pharmacy, China Medical University, Taichung, Taiwan, **5** Department of Life Science, Tunghai University, Taichung, Taiwan, **6** School of Medicine, National Yang-Ming University, Taipei, Taiwan

* s881056@gmail.com

## Abstract

### Introduction

Contrast-induced acute kidney injury (CI-AKI) is associated with high risks of morbidity and mortality. Hyperbilirubinemia might have some renal protection but with no clear cutoff value for protection. Related studies are typically on limited numbers of patients and only in conditions of vascular intervention.

### Methods

We performed this study to elucidate CI-AKI in patients after contrast-enhanced computed tomography (CCT). The outcomes were CI-AKI, dialysis and mortality. Patients were divided to three groups based on their serum levels of total bilirubin: ≤1.2 mg/dl, 1.3–2.0 mg/dl, and >2.0 mg/dl.

### Results

We enrolled a total of 9,496 patients who had received CCT. Patients with serum total bilirubin >2.0 mg/dl were associated with CI-AKI. Those undergoing dialysis had the highest incidence of PC-AKI (p<0.001). No difference was found between the two groups of total bilirubin ≤1.2 and 1.3–2.0 mg/dl. Patients with total bilirubin >2mg/dl were associated with CI-AKI (OR = 1.89, 1.53–2.33 of 95% CI), dialysis (OR = 1.40, 1.01–1.95 of 95% CI) and mortality (OR = 1.63, 1.38–1.93 of 95% CI) after adjusting for laboratory data and all comorbidities (i.e., cerebrovascular disease, coronary artery disease, peripheral arterial disease, and acute myocardial infarction, diabetes mellitus, hypertension, gastrointestinal bleeding, cirrhosis, peritonitis, ascites, hepatoma, shock lung and colon cancer). We concluded that total bilirubin level >2 mg/dl is an independent risk factor for CI-AKI, dialysis and mortality after CCT. These patients also had high risks for cirrhosis or hepatoma.

**Data Availability Statement:** All relevant data are within the manuscript and its Supporting Information files.

**Funding:** This study was funded by grant TCVGH-1063601B and TCVGH-1073604C from Taichung

Veterans General Hospital. The funders had no role in study design, data collection and analysis, decision to publish, or preparation of the manuscript.

**Competing interests:** The authors have declared that no competing interests exist.

## Conclusion

This is the first study providing evidence that hyperbilirubinemia (total bilirubin >2.0 mg/dl) being an independent risk factor for CI-AKI, dialysis and mortality after receiving CCT. Most patients with total bilirubin >2.0mg/dl had cirrhosis or hepatoma.

## Introduction

The nephrotoxicity of iodinated contrast media is well-known, and that is also the major cause of acute kidney injury (AKI). Formerly called contrast-induced nephropathy, it has now been called contrast-induced AKI (CI-AKI). The incidence of CI-AKI is high, with 12 to 50% morbidity [1–4]. Despite of its nephrotoxicity, the iodine-containing contrast medium is required to obtain good quality images. Therefore, its nephrotoxicity seems inevitable in clinical practice. Identified risk factors for CI-AKI are the following: impaired baseline renal function [5, 6], type and dose of contrast material [5, 7], conditions associated with reduced renal perfusion (such was heart failure [8], medications (angiotensin-converting enzyme inhibitor (ACEI) or angiotensin II receptor blockers (ARB), non-steroidal anti-inflammatory drug (NSAID), diuretics, and metformin [9]), volume depletion like diarrhea or vomiting and sepsis. Despite of such knowledge on risk factors and the preventive measure with volume expansion, the incidence of CI-AKI remains high. This indicates other undiscovered risk factors for CI-AKI.

Bile acid or bilirubin with low water solubility can cause cast formation within the low pH microenvironment of distal nephrons. Bile cast cholemic nephrosis or bile nephrosis has been reported since 1953 [10]. Later related studies were done in rabbits 1957 [11], on tubular injury in 1958 [12], and on autopsy [13]. Since 2000, only a few studies have been reported on renal toxicity related to bile acid or bilirubin [14–16]. Bilirubin is an endogenous circulating antioxidant with protective role on kidney damages [17]. Also, bilirubin has ability anti-inflammatory, complete inhibitory and lipid-lowering properties[18]. Patients with Gilbert's syndrome may experience mild jaundice and the condition was recently shown to reduce all-cause mortality by half [19]. Hypobilirubinemia was reported in 2012 to be a possible risk factor for end-stage kidney disease (ESRD), independent of the estimated glomerular filtration rate (eGFR) [20]. A recent study also supported the protective role of hyperbilirubinemia on renal functions (0.5±0.2 vs. 0.7±0.3 mg/dl, p<0.001) [21]. On the other hand, contradicting results were reported in another study as detailed below [22]. In that observational large hospital-based study of 2,678 adult outpatients, the total bilirubin was found to be inversely associated with eGFR in both non-diabetic (r = -0.17; p < 0.0001) and diabetic patients (r = -0.14; p < 0.05). Some issues remain controversial between serum bilirubin and renal function. First, there is no consensus regarding serum bilirubin in terms of its nephrotoxicity or renal protective role. Second, even if it is true for renal protection or nephrotoxicity, their effective cutoff values of hyperbilirubinemia remain unclear. Third, the effect of hyperbilirubinemia on CI-AKI was reported only in patients undergoing diagnostic angiography [23] or in coronary intervention [21] [24]. In the above clinical scenarios, the conditions are relatively simple. No studies have been done specifically on the relationship between serum bilirubin levels and CI-AKI after contrast-enhanced computed tomography (CCT). Fourth, the contrast volume, a possible risk factor for CI-AKI was not constant across studies. Finally, case numbers are not sufficiently large enough for adjusting confounding factors. Therefore, in the study, we determined the association between serum bilirubin and renal function after CCT (at a fixed volume of contrast volume, 100ml).

## Methods and materials

### Study design and patient population

We used in our hospital (Taichung Veterans General Hospital) a historical cohort that consisted data of 20,018 non-dialytic adult patients who had received the non-ionic iso-osmolar contrast medium, iodixanol (Visipaque, Chicago, IL, USA), for enhanced CT imaging during an approximately 15-year period (June 1, 2008 to March 31, 2015). The data recorded for each patient included a baseline serum level of creatinine and total bilirubin within two days before CCT. They were used to evaluate the association between serum bilirubin and renal outcome after CCT. Exclusion criteria were those with pre-existing AKI (defined according to KDIGO practice guideline[25]), with recent exposures to contrast media over the previous 30 days, volume of contrast medium not being to 100 ml (regular contrast volume for CCT), baseline serum levels of total bilirubin and creatinine within the two days before CCT not available, and post-contrast serum creatinine within one week after CCT not available.

Our study was approved by the institute review board of Taichung Veterans General Hospital approved this study (IRB TCVGH No:F15059). Patient informed consent was waived due to the pure data analysis nature of the study. There was no formal protocol for the prevention of contrast-induced nephropathy at this hospital over the study period.

### Baseline data retrieval and definition

The baseline stages of chronic kidney diseases (CKD) was calculated using the equation of modification of diet in renal disease (MDRD) equation [26]: eGFR (ml/min per 1.73 m$_2$) = 186*SCr$^{-1.154}$ *Age$^{-0.203}$*0.742 (if female). Medical records of patients were screened for all comorbid conditions: such as, cardiovascular and cerebrovascular disease (cerebrovascular attack, coronary artery disease, peripheral arterial disease, and acute myocardial infarction), metabolic disease (diabetes mellitus, and hypertension), gastrointestinal bleeding, liver disease (cirrhosis, peritonitis, ascites and hepatoma), shock and malignancy (lung and colon cancer). Medications screened were ACEi, ARB, NSAID, aspirin, aminoglycoside, loop diuretics, steroid, statin, and H$_2$-blocker (ranitidine and famotidine).

### Outcome data retrieval and outcome definitions

Primary outcome, PC-AKI, was defined according to the KDIGO practice guideline[25]: absolute increase of serum creatinine levels ≥0.3 mg/dl from baseline within 48 h, or ≥ 50% within 7 days after CCT [4]. Considering that urine volume had not been regularly collected, we therefore did not include the criterion of urine volume for AKI, as that was in the guideline of KDIGO. The secondary endpoint was the need of emergent hemodialysis within 30 days after CCT (as identified by the first recorded procedure of hemodialysis within 30 days after CCT). All the indication and timing for urgent hemodialysis for patients were the same as described below: refractory fluid overload even with diuretics, severe hyperkalemia (>6.5 meq/L) even after medication, >100 mg/dl of blood urea nitrogen, >6 mg/dl of serum creatinine, metabolic acidosis (<7.2), uremic encephalopathy, uremic bleeding and uremic pericarditis.

Patients were divided to three groups according to serum levels of total bilirubin: ≤1.2 mg/dl, 1.3~2.0 mg/dl, and >2.0 mg/dl. The first cutoff value was 1.2 mg/dl which just exceeded the normal range. Also, the Youden index was used to determine the cutoff value to predict AKI was set to >1.2 mg/dl of total bilirubin within 30 days after CCT (S1 Fig). The other cutoff value of total bilirubin> 2.0 mg/dl followed the Child–Pugh classification[27]. The incidence of AKI in patients with total bilirubin > 2.0 mg/dl was also classified into different three liver conditions (cirrhosis, hepatoma and no cirrhosis or hepatoma) in S2 Table. Furthermore, the

incidence of AKI in patients without any liver conditions was classified according to serum levels of total bilirubin in S3 Table.

## Statistical analyses

Quantitative data were expressed as mean ± standard deviation. Nominal and categorical variables were compared using the *Chi*-square likelihood ratio or Fisher exact test with bonferroni post-hoc analyses to detect differences between data pairs. Continuous variables were compared using the nonparametric Wilcoxon test. The stepwise multivariate logistic regression analysis was used to examine the independent association of PC-AKI with patient-related characteristics and comorbidities. In model 1, we adjusted for all comorbidities. They included cerebrovascular disease, coronary artery disease, peripheral arterial disease, and acute myocardial infarction, diabetes mellitus, hypertension, gastrointestinal bleeding, cirrhosis, peritonitis, ascites, hepatoma, shock lung and colon cancer. In model 2, in addition to adjusting for the above comorbidities, we further adjusted for the following: stage of CKD, hemoglobin, serum sodium, serum potassium, prothrombin time, international normalized ratio, the usage of aspirin, aminoglycoside, loop diuretics, ACEi, ARB, NSAID, and the use of fluid replacement >1 liter on the day of CCT. The associations between serum bilirubin level (>2.0 mg/dl) and the characteristics and risks of PC-AKI, and dialysis within 30 days after CCT were calculated by odds ratio (OR) and 95% confidence interval (CI). A two-sided p value of <0.05 represented statistical significance. The SPSS software (Statistical Package for the Social Science, version 20.0, Armonk, NY, USA) was used for statistical analyses.

## Results

Initially, we recruited a total 20,018 who received CCT patients for this study. After exclusion, a total of 9,496 patients of these patients without missing data were enrolled in the final study cohort (Fig 1). Their baseline characteristics are list in Table 1, according to three categories based on serum levels of total bilirubin (ie. ≤1.2, 1.3~2.0, and >2.0 mg/dl). Patients with higher levels of total bilirubin had the following characteristics: older (p = 0.012), more males (p<0.001), lower serum albumin levels (p<0.001), more with hyponatremia (p<0.001), longer prothrombin time (p<0.001), more metabolic acidosis (p<0.001), fewer cerebrovascular attacks (p<0.001), more cirrhosis (p<0.001), more hepatoma (p<0.001), fewer lung cancer (p<0.001), more shocks (p<0.001), more peritonitis (p<0.001), more ascites (p<0.001), and more gastrointestinal bleeding (p<0.001). Patients with higher total bilirubin also received the medications as follow: less NSAIDs (p<0.001), more aspirin (p<0.001), more aminoglycosides (p<0.001), more loop diuretics (p<0.001), less ARB (p = 0.003), less steroid (p<0.001), less statin (p<0.001). They also received more fluid replacement (>1 liter) (p = 0.009). Higher total bilirubin was associated with more PC-AKI (p<0.001) and more incidence of urgent dialysis (p<0.001).

Patients in the group with total bilirubin>2mg/dl had more AKI (16.4% vs. 9.3% vs. 7.9%) compared to the other two groups with lower levels (≤1.2 mg/dl (p<0.001) and 1.3~2.0 mg/dl (p<0.001))(Fig 2A). Similarly, patients with total bilirubin>2mg/dl were prone to receive urgent hemodialysis compared to the other two groups of patients (≤1.2 mg/dl (p<0.001) or 1.3~2.0 mg/dl (p<0.001)) (Fig 2B). No difference was found between the two groups with lower total bilirubin levels regarding AKI or the incidence of urgent dialysis.

Total bilirubin>2mg/dl was associated with AKI (OR = 2.05, 1.72–2.45 of 95% CI), urgent dialysis (OR = 1.52, 1.14–2.02 of 95% CI) and mortality (OR = 1.82, 1.59–2.08 of 95% CI) (Table 2) after adjusting all comorbidities in model 1 (Table 1). After further adjusting for comorbidities, medications and laboratory data in model 2 (Table 1), total bilirubin >2 mg/dl

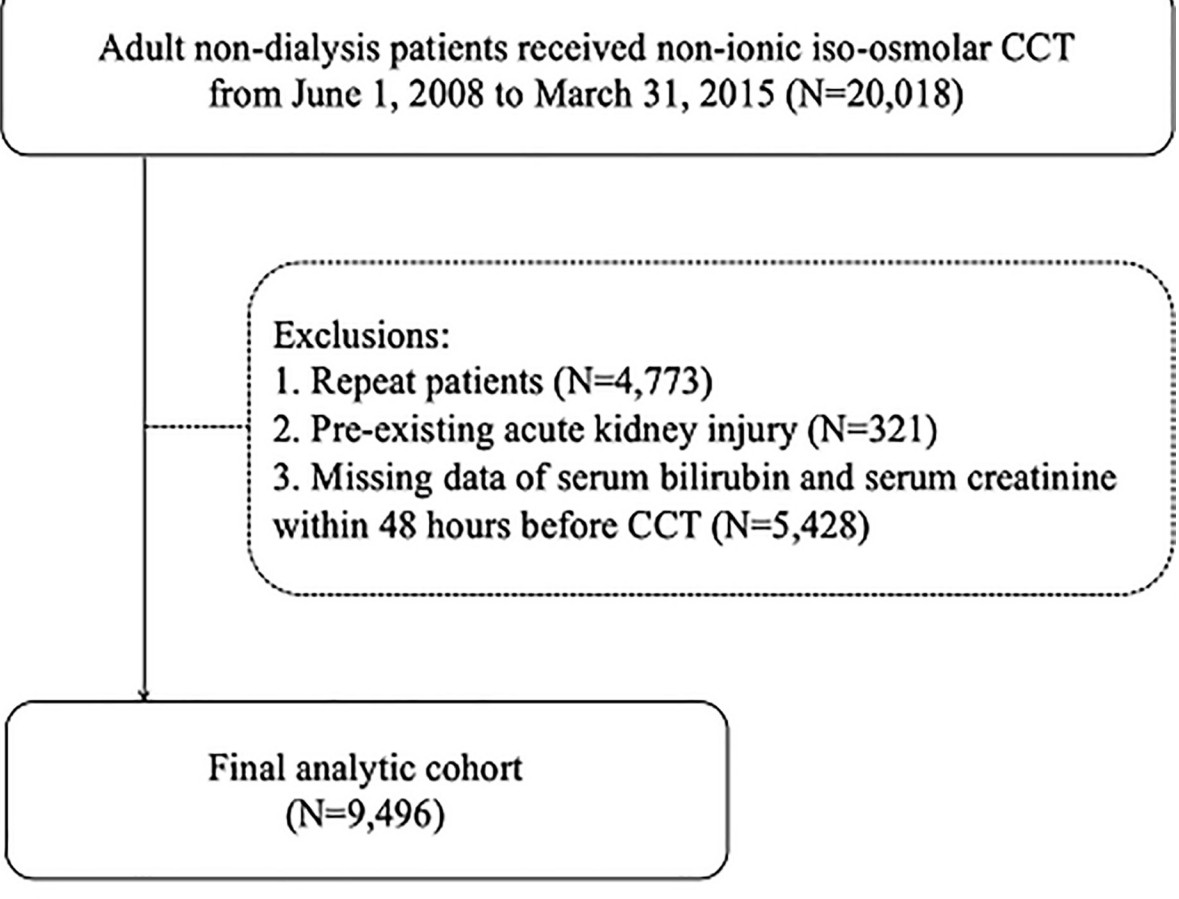

Adult non-dialysis patients received non-ionic iso-osmolar CCT from June 1, 2008 to March 31, 2015 (N=20,018)

Exclusions:
1. Repeat patients (N=4,773)
2. Pre-existing acute kidney injury (N=321)
3. Missing data of serum bilirubin and serum creatinine within 48 hours before CCT (N=5,428)

Final analytic cohort
(N=9,496)

**Fig 1. Algorithm of patient selection.**

was still associated with more AKI (OR = 1.89, 1.53–2.33 of 95% CI), dialysis (OR = 1.40, 1.01–1.95 of 95% CI) and mortality (OR = 1.63, 1.38–1.93 of 95% CI) in all patients (Table 2). Therefore, total bilirubin >2 mg/dl is an independent risk factor for AKI, dialysis and mortality after CCT in this cohort.

Compared to no cirrhosis or hepatoma, either with cirrhosis or hepatoma, patients are prone to have total bilirubin>2.0mg/dl (34% vs. 10%, p<0.001; 27.8% vs. 10%, p<0.001; respectively) (Fig 3). Because total bilirubin> 2 mg/dl is more common in patients with liver conditions, we further analyzed the association between total bilirubin >2 mg/dl and renal outcome in patients with baseline liver diseases (Table 3). The total bilirubin> 2 mg/dl was still associated with more AKI (OR = 3.50, 2.14–5.72 of 95% CI), and mortality (OR = 2.21, 1.58–3.11 of 95% CI) in patients with cirrhosis. Similarly, total bilirubin >2 mg/dl was also associated with more AKI (OR = 3.24, 1.89–5.56 of 95% CI), and mortality (OR = 2.41, 1.65–3.50 of 95% CI) in patients with hepatoma (Table 3). The ORs for AKI, dialysis and mortality are more significant in the cirrhosis group (3.50 vs. 1.89, 1.55 vs. 1.40; 2.21 vs. 1.63) (Table 3) or the hepatoma group (3.24 vs. 1.89; 1.35 vs. 1.40; 2.41 vs. 1.63) (Table 3) than in whole patients group (Table 2). In summary, patients with worse liver conditions (total bilirubin>2 mg/dl) in baseline cirrhosis or hepatoma (Table 3) had higher and additional risk of worse renal outcomes than whole population (Table 2).

**Table 1. Baseline characteristics of patients.**

| Total bilirubin | ≤1.2 (mg/dl) | 1.3–2.0 (mg/dl) | >2.0 (mg/dl) | All | *P* value |
|---|---|---|---|---|---|
| Numbers | 7173 | 995 | 1368 | 9496 | |
| Age (years) | 64.63±16.31 | 62.62±16.66 | 65.88±15.89 | 64.91±16.29 | 0.012 |
| ≧65 years | 3783 (52.7%) | 527 (55.2%) | 765 (55.9%) | 5075 (53.4%) | 0.051 |
| Female | 2818 (39.3%) | 301(31.5%) | 415(30.3%) | 3534 (37.2%) | <0.001 |
| **Stages of CKD** | | | | | <0.001 |
| 1 | 2730 (38.1%) | 350 (36.6%) | 541 (39.5%) | 3621 (38.1%) | |
| 2 | 2292 (32.0%) | 306 (32.0%) | 371 (27.1%) | 2969 (31.3%) | |
| 3a | 866 (12.1%) | 140 (14.7%) | 176 (12.9) | 1182 (12.4%) | |
| 3b | 651 (9.1%) | 91 (9.5%) | 129 (9.4%) | 871 (9.2%) | |
| 4 | 381 (5.3%) | 49(5.1%) | 106(7.7%) | 536(5.6%) | |
| 5 | 253(3.5%) | 19(2.0%) | 45(3.3%) | 317(3.3%) | |
| **Laboratory data of blood** | | | | | |
| Hemoglobin (g/dl) | 12.14±2.53 | 12.60±2.69 | 12.02±2.76 | 12.17±2.59 | <0.001 |
| Albumin (g/dl) | 3.47±0.72 | 3.29±0.75 | 3.12±0.75 | 3.40±0.74 | <0.001 |
| Calcium (mg/dl) | 8.04±1.63 | 7.84±1.66 | 7.88±1.54 | 8.00±1.62 | <0.001 |
| Sodium (meq/L) | 137.96±5.60 | 136.98±6.35 | 136.00±6.07 | 137.58±5.79 | <0.001 |
| Potassium (mg/dl) | 4.09±0.70 | 4.03±0.78 | 4.06±0.81 | 4.08±0.72 | 0.043 |
| Uric acid (mg/dl) | 6.59±2.56 | 7.72±3.83 | 6.70±3.14 | 6.68±2.74 | 0.043 |
| Prothrombin time (s) | 11.48±4.78 | 12.63±4.61 | 14.18±7.20 | 12.02±5.30 | <0.001 |
| pH | 6.87±0.83 | 6.99±0.77 | 6.95±0.78 | 6.90±0.81 | <0.001 |
| $HCO_3^-$ (mmo/L) | 23.96±5.28 | 23.40±5.13 | 22.81±5.03 | 23.69±5.23 | <0.001 |
| **Comorbidity** | | | | | |
| Diabetes mellitus | 2116 (38.1%) | 276 (28.9%) | 422 (30.8%) | 2814 (29.6%) | 0.529 |
| Hypertension | 3529 (49.2%) | 469(49.1%) | 606 (44.3%) | 4604 (48.5%) | 0.004 |
| Cerebrovascular attack | 1114 (15.5%) | 139 (14.6%) | 132(9.6%) | 1385 (14.6%) | <0.001 |
| Peripheral arterial disease | 178 (2.5%) | 16 (1.7%) | 28 (2.0%) | 222 (2.3%) | 0.224 |
| Cirrhosis | 560 (7.8%) | 208 (21.8%) | 459 (33.6%) | 1227 (12.9%) | <0.001 |
| Hepatoma | 501 (7.0%) | 152 (15.9%) | 358 (26.2%) | 1011 (10.6%) | <0.001 |
| Colon cancer | 796 (11.1%) | 74 (7.7%) | 101 (7.4%) | 971 (10.2%) | <0.001 |
| Lung cancer | 1466 (20.4%) | 110 (11.5%) | 139 (10.2%) | 1715 (18.1%) | <0.001 |
| Atrial fibrillation | 630 (8.8%) | 124 (13.0%) | 130 (10.2%) | 884 (9.3%) | <0.001 |
| Coronary arterial disease | 1179 (16.4%) | 184 (19.3%) | 204 (14.9%) | 1567 (4.6%) | 0.020 |
| Myocardial infarction | 346 (4.8%) | 34 (3.6%) | 58 (4.2%) | 438 (4.6%) | 0.168 |
| Shock | 123 (1.7%) | 24 (2.5%) | 55 (4.0%) | 202 (2.1%) | <0.001 |
| Peritonitis | 128 (1.8%) | 32 (3.4%) | 72 (5.3%) | 232 (2.4%) | <0.001 |
| Ascites | 70 (1.0%) | 20 (2.1%) | 66 (4.8%) | 156 (1.6%) | <0.001 |
| Gastrointestinal bleeding | 355 (4.9%) | 65 (6.8%) | 113(8.3%) | 533(5.6%) | <0.001 |
| **Medication** | | | | | |
| Non-steroidal anti-inflammatory drugs | 3666 (51.1%) | 391 (40.9%) | 503 (36.8%) | 4560 (48.0%) | <0.001 |
| Aspirin | 1393 (19.4%) | 180 (18.8%) | 192 (14.0%) | 1765 (18.6%) | <0.001 |
| Aminoglycoside | 3092 (43.1%) | 456 (47.7%) | 686 (50.1%) | 4234 (44.6%) | <0.001 |
| Loop diuretics | 3866 (53.9%) | 585 (61.3%) | 888 (64.9%) | 5339 (56.2%) | <0.001 |
| Angiotensin-converting-enzyme inhibitor | 643 (9.0%) | 97 (10.2%) | 120 (8.8%) | 860 (9.1%) | 0.446 |
| Angiotensin receptor blockers | 1466 (20.4%) | 178 (18.6%) | 227 (16.6%) | 1871 (19.7%) | 0.003 |
| Steroid | 1561 (21.8%) | 155 (16.2%) | 179 (13.1%) | 1895 (20.2%) | <0.001 |
| Statin | 711 (9.9%) | 77 (8.1%) | 76 (5.6%) | 864 (9.1%) | <0.001 |
| Ranitidine | 654 (9.1%) | 71 (7.4%) | 116 (8.5%) | 841 (8.9%) | 0.198 |

*(Continued)*

**Table 1.** (Continued)

| Total bilirubin | ≤1.2 (mg/dl) | 1.3–2.0 (mg/dl) | >2.0 (mg/dl) | All | *P* value |
|---|---|---|---|---|---|
| Famotidine | 1147 (16.0%) | 149 (15.6%) | 223 (16.3%) | 1519 (16.0%) | 0.902 |
| Fluid replacement > 1000c.c. | 1426 (19.9%) | 199 (20.8%) | 322 (23.5%) | 1947 (20.5%) | 0.009 |
| Acute kidney injury | 569 (7.9%) | 89 (9.3%) | 225 (16.4%) | 883 (9.3%) | <0.001 |
| Dialysis within 30 days | 304 (4.2%) | 47 (4.9%) | 98 (7.2%) | 449 (4.7%) | <0.001 |

Chi-square test and one-way ANOVA.

The incidence of AKI in total bilirubin> 2 mg/dl according to different baseline liver conditions were shown in Fig 4. In patients with cirrhosis (Fig 4A), hepatoma (Fig 4B), or without cirrhosis or hepatoma (Fig 4C), patients with total bilirubin >2.0mg/dl also had more CI-AKI than patients lower levels of total bilirubin (p<0.001). This result also indicated that total bilirubin > 2 mg/dl was independent risk factors for CI-AKI in different baseline liver conditions. In patients with total bilirubin> 2 mg/dl (n = 1368), baseline characteristics were shown in S2 Table according to baseline liver conditions. Moreover, patients with total bilirubin > 2 mg/dl but without cirrhosis or hepatoma (n = 7826) were also shown S3 Table. Non-cirrhosis or hepatoma related hyperbilirubinemia (>2 mg/dl) was associated with older age (p<0.001), hypoalbuminemia (p<0.001), longer prothrombin time (p<0.001), more metabolic acidosis (p<0.001), less lung cancer (p<0.001), less cerebrovascular attack (p<0.001), less NSAID (p<0.001) and steroid usage (p<0.001), more aminoglycoside usage (p<0.001), and more fluid replacement (p = 0.017).

Post-CCT mortality is shown in Fig 5. Patients with total bilirubin levels ≤2.0 mg/dl and without AKI had the lowest mortality. For patients with either total bilirubin >2.0mg/dl or

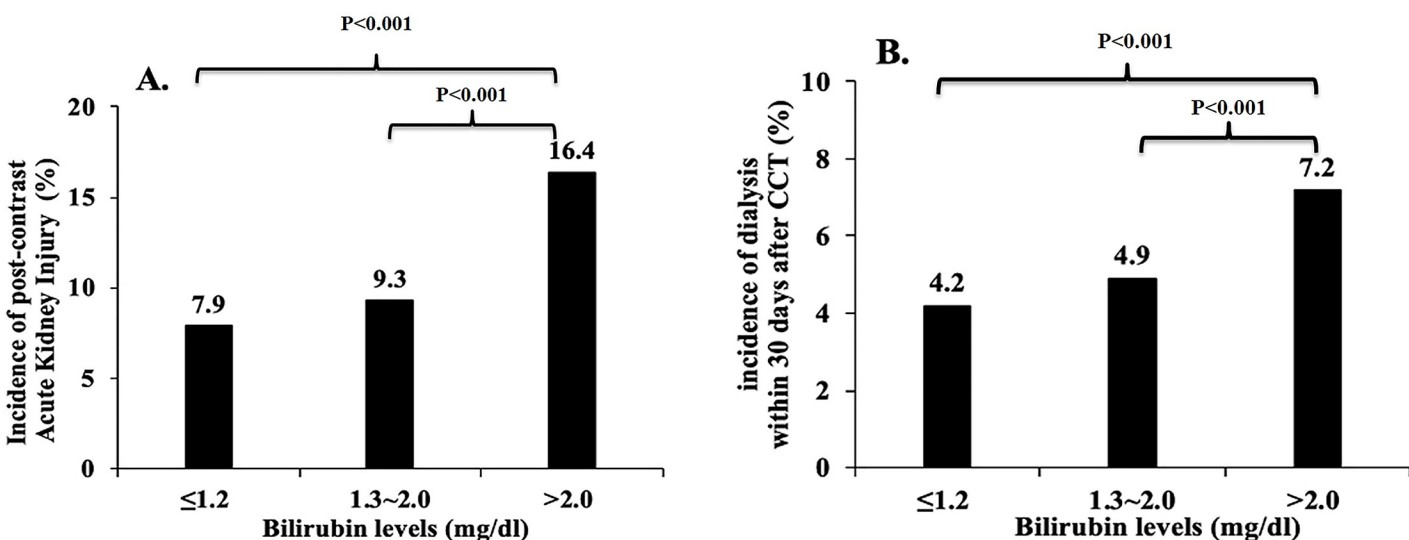

**Fig 2. Renal outcomes after contrast-enhanced computed tomography in all patients divided by different serum levels of bilirubin.** 2A. Incidence of **acute kidney after** contrast-enhanced computed tomography (CCT) divided by different serum levels of total bilirubin. No statistical significance was noticed between the groups of ≤1.2 mg/dl and 1.3–2.0 mg/dl of serum bilirubin (7.9 vs. 6.3%). Differences are statistically significant between groups of ≤1.2 mg/dl and >2.0 mg/dl of serum bilirubin (7.9 vs. 16.4%, p<0.001); and between 1.3–2.0 mg/dl and >2.0 mg/dl of serum bilirubin (9.3 vs. 16.4%, p<0.001). 2B. Incidence of **urgent dialysis** within 30 days after contrast-enhanced computed tomography (CCT) divided by different serum levels of total bilirubin. No statistical significance was found between the groups of ≤1.2 mg/dl and 1.3–2.0 mg/dl of serum bilirubin (4.2 vs. 4.9%). Differences are statistically significant between groups of ≤1.2 mg/dl and >2.0 mg/dl of serum bilirubin (4.2 vs. 7.2%, p<0.001); and between 1.3–2.0 mg/dl and >2.0 mg/dl of serum bilirubin (4.9 vs. 7.2%, p<0.001).

**Table 2. Association between serum bilirubin > 2.0 mg/dL and the risk of acute kidney injury, dialysis within 30 days, and death after contrast-enhanced computerized tomography (CCT).**

|  | Odds Ratio | 95%CI | P value |
|---|---|---|---|
| Risk of acute kidney injury after CCT |  |  |  |
| Unadjusted | 2.23 | (1.90–2.63) | <0.001** |
| Adjusted, model 1 | 2.05 | (1.72–2.45) | <0.001** |
| Adjusted, model 2 | 1.89 | (1.53–2.33) | <0.001** |
| Risk of dialysis within 30 days after CCT |  |  |  |
| Unadjusted | 1.71 | (1.36–2.16) | <0.001** |
| Adjusted, model 1 | 1.52 | (1.14–2.02) | 0.005** |
| Adjusted, model 2 | 1.40 | (1.01–1.95) | 0.044* |
| Risk of death after CCT |  |  |  |
| Unadjusted | 1.98 | (1.76–2.24) | <0.001** |
| Adjusted, model 1 | 1.82 | (1.59–2.08) | <0.001** |
| Adjusted, model 2 | 1.63 | (1.38–1.93) | <0.001** |

Definition of acute kidney injury is an absolute increment of serum creatinine ≥0.3 mg/dl from baseline within 48 hours or ≥50% within 7 days after contrast-enhanced computerized tomography (CCT). Model 1, adjusted for the comorbidities listed in Table 1. Model 2, adjusted for the stage of CKD, hemoglobin, serum sodium, serum potassium, prothrombin time, international normalized ratio, the usage of aspirin, aminoglycoside, loop diuretics, ACE inhibitors/ARB, non-steroidal anti-inflammatory drugs, the use of fluid replacement > 1 liter on the day of CCT, plus covariates listed in Model 1.

*: $p < 0.05$

**: $p < 0.01$.

with AKI, mortality was higher (15% vs. 5.9%, p<0.001; 26.7% vs. 5.9%, p<0.001). Patients with simultaneously AKI and total bilirubin >2.0mg/dl had the highest mortality (41.8% vs. 5.9%, p<0.001).

## Discussion

We conduct this study to elucidate the association between serum total bilirubin and CI-AKI in patients undergoing CCT. Previous studies on this issue only involved in patients undergoing angiography for coronary [21] [24] or with peripheral arterial intervention [23]. Our study is the first on patients undergoing CCT and it has largest of cases focusing on the association between total bilirubin and renal function. Previous studies were focused on relatively simple condition likely due to small numbers of cases and hard to adjust for confounding factors. Patients underwent CCT have more complicated comorbidities (e.g. shock, bleeding, cirrhosis, and hepatoma) and therefore more cases are required to analyze common clinical scenarios of CI-AKI after CCT than those after vascular intervention. Negative or inconclusive results on the relationship between bilirubin and renal functions were observed mostly in patients with multiple comorbidities such as hemorrheological disorders, infectious diseases, and decompensated heart failure. These conditions confound the effect of bilirubin on prognosis and the results obtained should be interpreted with caution. The importance of this study is the large number of cases (>9,000) so that all comorbidities and any other confounding factors could be adjusted for.

Mild elevation of serum bilirubin has renal protective funtion related to antioxidative effects of bilirubin [17–19, 21, 28–34]. Bilirubin can bind to albumin [17] and exhibits protein anitoxidative, anti-inflammatory, complete inhibitory and lipid-lowering properties [18]. It may also protect against all-cause mortality and CV diseases. For patients of Gilbert's

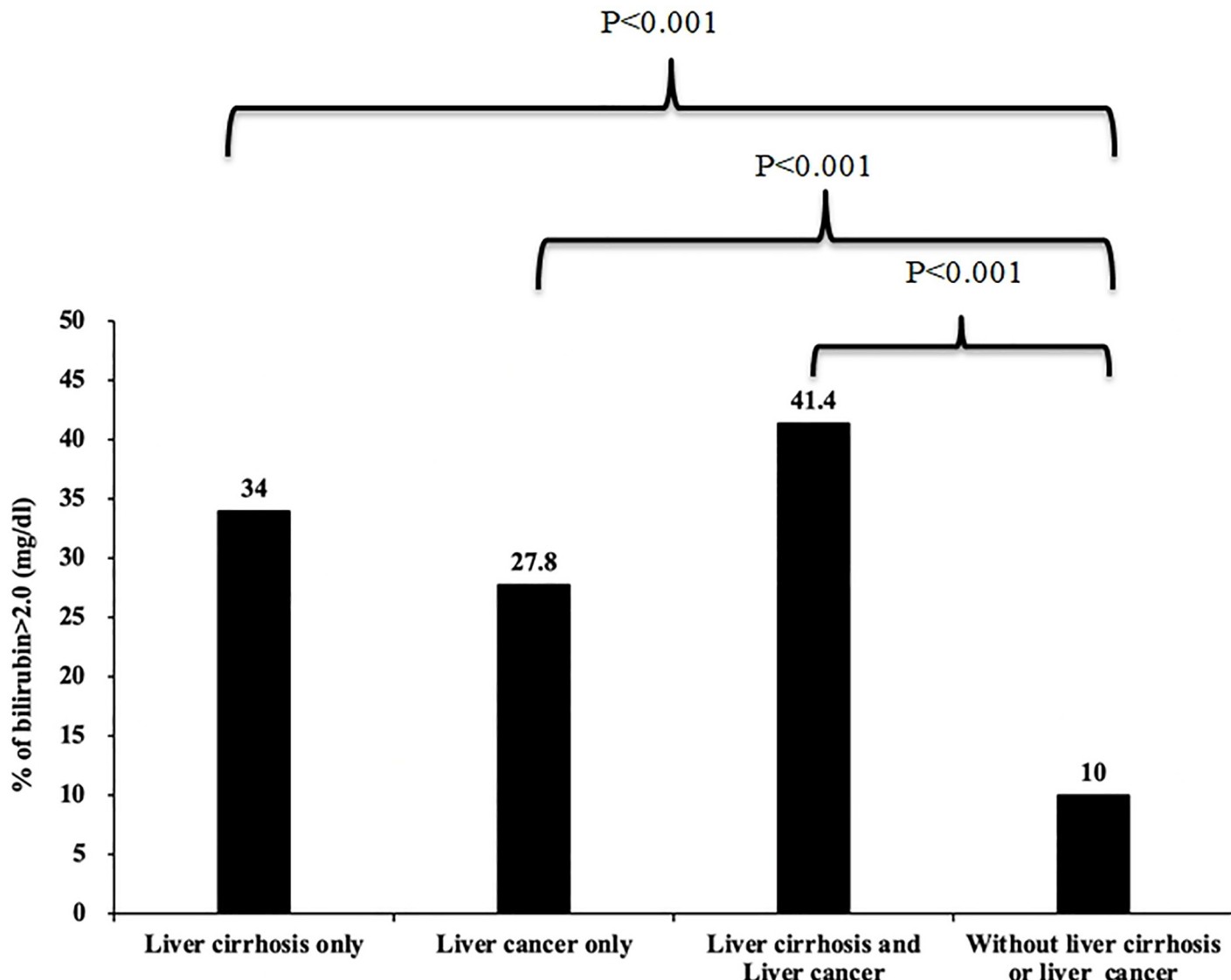

**Fig 3. Incidence of serum bilirubin > 2.0 mg/dl in patients with different liver conditions.** Differences are statistically significant between groups of serum bilirubin > 2.0mg/dl in patients with cirrhosis and without liver conditions (34 vs. 10%, p<0.001), and between groups of patients with hepatoma and without liver conditions (27.8 vs. 10%, p<0.001).

syndrome with mild jaundice the incidence of all-cause mortality is lowerd by half [19]. In a Korean study on a cohort (n = 1,458) with primary IgA nephropathy [28], quartile levels of bilirubin of patients were divided into: <0.4 mg/dL, 0.4–0.5 mg/dL, 0.6–0.7 mg/dL, and >0.8 mg/d. They found that the level of bilirubin was negatively associated with the incidence of ESRD. In another Korean study (1,363 patients) [29], total serum bilirubin was positively associated with eGFR and negatively associated with proteinuria. But their highest level of total bilirubin was only 0.77±0.22 mg/dl. In 2012, 544 patients receiving coronary intervention experienced fewer CI-AKI with higher serum levels of bilirubin [21]. The dovodomg values of bilirubin were ≤0.5 mg/dl, 0.5–0.7 mg/dl, and >0.7 mg/dl. Another study druing the same year showed that hypobilirubinemia is a possible risk factor of ESRD [20]. Their dividing values of bilirubin were <0.55, 0.59, 0.56, 0.47, and 0.36 mg/dl [20]. Bilirubin could attenuate cyclosporine-induced nephropathy of tubular injury by inhibiting oxidative stress and

**Table 3. Association between serum bilirubin > 2.0 mg/dl and the risk of acute kidney injury, dialysis within 30 days and death after contrast-enhanced computerized tomography (CCT) in patients with liver cirrhosis and liver cancer.**

| | Odds Ratio | 95%CI | | P value |
|---|---|---|---|---|
| Patients with **liver cirrhosis** and serum bilirubin > 2.0 mg/dl | | | | |
| Risk of acute kidney injury after CCT | | | | |
| Unadjusted | 3.55 | (2.47- | 5.09) | <0.001** |
| Adjusted, model 1 | 3.69 | (2.50- | 5.44) | <0.001** |
| Adjusted, model 2 | 3.50 | (2.14- | 5.72) | <0.001** |
| Risk of dialysis within 30 days after CCT | | | | |
| Unadjusted | 1.48 | (0.94- | 2.33) | 0.093 |
| Adjusted, model 1 | 1.58 | (0.91- | 2.75) | 0.103 |
| Adjusted, model 2 | 1.55 | (0.77- | 3.11) | 0.220 |
| Risk of death within 30 days after CCT | | | | |
| Unadjusted | 2.29 | (1.80- | 2.91) | <0.001** |
| Adjusted, model 1 | 2.36 | (1.81- | 3.08) | <0.001** |
| Adjusted, model 2 | 2.21 | (1.58- | 3.11) | <0.001** |
| Patients with **liver cancer** and serum bilirubin > 2.0 mg/dl | | | | |
| Risk of acute kidney injury after CCT | | | | |
| Unadjusted | 3.50 | (2.36- | 5.18) | <0.001** |
| Adjusted, model 1 | 3.47 | (2.26- | 5.33) | <0.001** |
| Adjusted, model 2 | 3.24 | (1.89- | 5.56) | <0.001** |
| Risk of dialysis within 30 days after CCT | | | | |
| Unadjusted | 1.37 | (0.76- | 2.48) | 0.296 |
| Adjusted, model 1 | 1.33 | (0.67- | 2.66) | 0.413 |
| Adjusted, model 2 | 1.35 | (0.52- | 3.52) | 0.536 |
| Risk of death within 30 days after CCT | | | | |
| Unadjusted | 2.33 | (1.79- | 3.03) | <0.001** |
| Adjusted, model 1 | 2.26 | (1.70- | 3.00) | <0.001** |
| Adjusted, model 2 | 2.41 | (1.65- | 3.50) | <0.001** |

Definition of acute kidney injury is an absolute increment of serum creatinine $\geq$0.3 mg/dl from baseline within 48 hours or $\geq$50% within 7 days after contrast-enhanced computerized tomography (CCT). Model 1, adjusted for the comorbidities listed in Table 1. Model 2, adjusted for the stage of CKD, hemoglobin, serum sodium, serum potassium, prothrombin time, international normalized ratio, the usage of aspirin, aminoglycoside, loop diuretics, ACE inhibitors/ARB, non-steroidal anti-inflammatory drugs, the use of fluid replacement > 1 liter on the day of CCT, plus covariates listed in Model 1.

*: $p$ <0.05

**: $p$ <0.01

apoptosis in HK-2 cells [33]. With higher levels of bilirubin, the greater suppression on oxidative stress might attenuate the progression of CKD [17, 34]. Importantly, all the above clinical studies are in support of renal protection by bilirubin were based on mild elevation of bilirubin levels oftern within normal range (<1.2 mg/dl). That is why one of cutoff levels of bilirbuin in our study is 1.2 mg/dl.

However, not all studies supported the same conclusion [14, 22, 35–42]. For example, Targher et al[22], did an observational study with 2,678 patients and they found that total bilirubin was inversely associated with eGFR in both non-diabetic (r = -0.17; p<0.0001) and diabetic patients (r = -0.14; p<0.05). In another study, the increased total bilirubin levels are independently associated with decreasing eGFR [35]. The authors hypothesized that the discrepancy in conclusion with the mainstream literature maybe due to nonalcoholic fatty liver disease [35]. Similarly, the benefits of hyperbilirubinemia might be confounded by other factors like: cholemic nephrosis [14, 38], cholestasis [39], infection with malaria [40, 41] and spontaneous

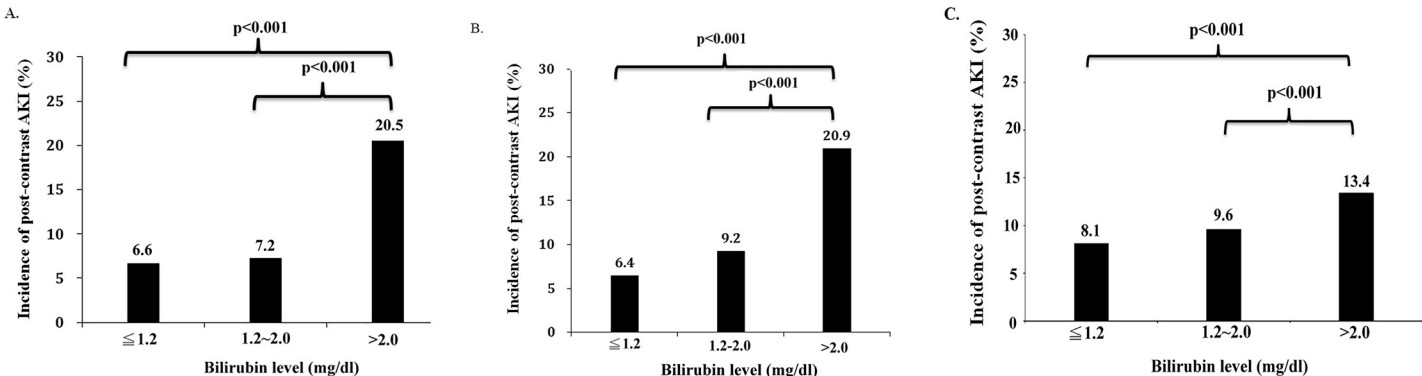

**Fig 4. Acute kidney injury after contrast-enhanced computed tomography (CCT) in patients with different liver conditions divided by different serum levels of total bilirubin.** 4A. Incidence of **acute kidney injury** after contrast-enhanced computed tomography (CCT) divided by different serum levels of total bilirubin in patients with **cirrhosis**. No statistical significance was noticed between the groups of ≤ 1.2 mg/dl and 1.3–2.0 mg/dl of serum bilirubin (6.6 vs. 7.2%). Differences are statistically significant between groups of ≤ 1.2 mg/dl and > 2.0 mg/dl of serum bilirubin (6.6 vs. 20.5%, p<0.001), and between 1.3–2.0 mg/dl and >2.0 mg/dl of serum bilirubin (7.2 vs. 20.5%, p<0.001). 4B. Incidence of **acute kidney injury** after contrast-enhanced computed tomography (CCT) divided by different serum levels of total bilirubin in patients with **hepatoma**. No statistical significance was noticed between the groups of ≤1.2 mg/dl and 1.3–2.0 mg/dl of serum bilirubin (6.4% vs. 9.2%). Differences are statistically significant between groups of ≤ 1.2 mg/dl and > 2.0 mg/dl of serum bilirubin (6.4 vs. 20.9%, p<0.001), and between 1.3–2.0 mg/dl and >2.0 mg/dl of serum bilirubin (9.2 vs. 20.9%, p<0.001). 4C. Incidence of **acute kidney injury** after contrast-enhanced computed tomography (CCT) divided by different serum levels of total bilirubin in patients without **hepatoma or cirrhosis**. No statistical significance was noticed between the groups of ≤1.2 mg/dl and 1.3–2.0 mg/dl of serum bilirubin (8.1% vs. 9.6%). Differences are statistically significant between groups of ≤ 1.2 mg/dl and > 2.0 mg/dl of serum bilirubin (8.1 vs. 13.4%, p<0.001), and between 1.3–2.0 mg/dl and >2.0 mg/dl of serum bilirubin (9.6 vs. 13.4%, p<0.001).

bacterial peritonitis [36], and heart failure [42]. On the contrary, in patients with non-fulminant hepatitis A, lower levels of bilirubin are associated with fewer [37]. In the vent of hepatopathy, total bilirubin level is no longer associated with the protective effects from cardiovascular disease. Such findings highlighted the problematic use of total bilirubin measures

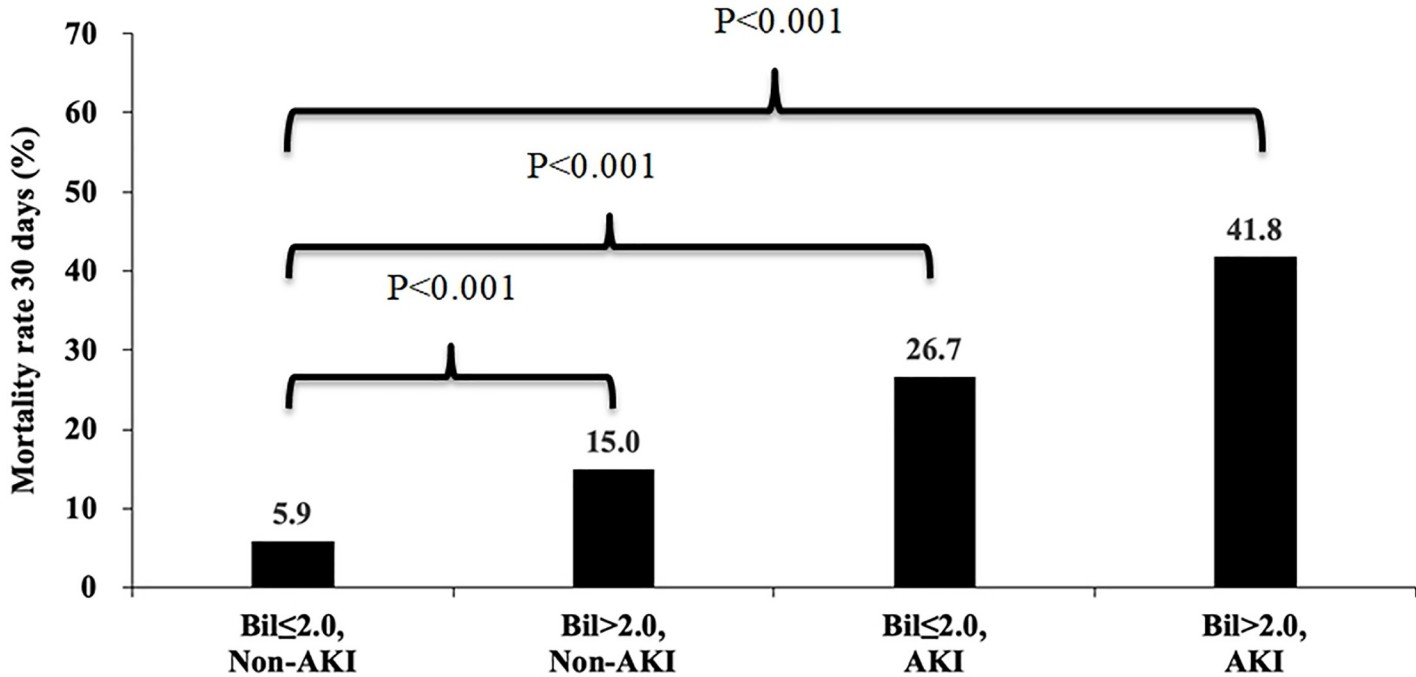

**Fig 5. Compared to no AKI and ≤ 2.0 mg/dl of serum bilirubin, patient mortality within 30 days after contrast-enhanced computed tomography (CCT) increased if AKI had occurred (26.7%, p<0.001) or > 2.0 mg/dl of serum bilirubin (15.0%, p<0.001).** Patients with both AKI and > 2.0 mg/dl of serum bilirubin had the highest risk of mortality (41.8%, p<0001).

in any investigation of the protective effects of bilirubin [43]. These studies with negative findings also indicated that patients with severe liver disease are more likely to develop hepatic encephalopathy and hepatorenal syndrome. That result is consistent with ours in that total bilirubin level >2 mg/dl was mostly in patients with cirrhosis or hepatoma and total bilirubin >2.0mg/dl is associated with CI-AKI after CCT.

Some new information has emerged from this study. First, with abnormally severe hyperbilirubinemia (>1.2 mg/dl), we found that the higher the total bilirubin, the more CI-AKI after CCT, especially > 2.0 mg/dl of total bilirubin. The renal protection by mild elevations of total bilirubin was all < 1.2 mg/dl, or within the normal range. Second, the mild elevation of total bilirubin maybe contributed by unconjugated bilirubin rather than by conjugated bilirubin. In patients with total bilirubin > 2.0mg/dl, mostly due to "patological" hyperbilibubinemia related to cirrhosis or hepatoma. Third, 10% patients without cirrhosis or hepatoma still had their total bilirubin > 2.0mg/dl, and they still had more CI-AKI after CCT. Total bilirubin > 2.0mg/dl is therefore an independent risk factor for CI-AKI, which indicated an association between CI-AKI and total bilirubin level > 2.0mg/dl.

The inciden PC-AKI in patients with total bilirubin> 2.0 mg/dl was 16.4%. This is the frist sutdy to point out the incidence of PC-AKI regarding hyperbilirubinemia. The incidence of PC-AKI varied from 12 to 50% according to different baseline conditions[2–4, 44–46]. The incidence of PC-AKI in patients with total bilirubin> 2.0 mg/dl maybe relatively low because of the following reasons. First, we recruited both inpatients and outpatients. Overall, this population was all patients receiveing CCT with relatively mild disease. Most patients had good baseline renal function (66.6% with baseline eGFR>60 ml/min.1.732m$^2$) and without anemia (12.02 g/dl of mean Hb). Only 30.8% patients had DM and shock was only noticed in 4.0% patients. Second, 33.6% patients had cirrhosis and 26.2% patients had hepatoma. Patients with liver disease may have less skeletal muscle mass, which made less creatine storage, followed by less conversion of creatine to creatinine[47]. Creatinine-based formula may underestiate renal dysfunction[47]. Finally, in this study, we point out the importance of the incidence of PC-AKI in patients with total bilirubin>2.0 mg/dl. Therefore, without previous studies for comparison, the incidence (16.4%) cannot be considered as too high or too low.

CI-AKI is primarily a condition of glomerular hypotension related to vasoconstriction [48]. Biliriun had the ability to neuralize reactive oxidative species(ROS) [49]. Increased renal ROS leads renin releases, followed by more secretino of angiotenin II [50]. ROS's effect on efferent vasocontriction is reduced[48], causing AKI by the reduced intra-glomeurlar pressure. In an animal study, moderate hyperbilirubinemia prevents angiotenin II-dependent hypertension by decreasing vascular oxidative stress [51]. The blocked angiotensin II may also reduce vasoconstrction over efferent arterioles [48]. Moreover, hyperbilirubinemia may also lower both artery pulse pressure [52] and intraglomerula pressure[17]. Therefore, severe hyperbilirubinemia may lower pre-renal perfusion by decreassed intraglomerrular perssure, predisposing CI-AKI. Another cause could be direct tubular damages by biliruin. Followed by the condition of pre-renal AKI, CI-AKI is a result of due to tubular injuries. The association between hyperbilirubinemia and AKI in the condition of liver cirrhosis might be partially explained by a direct cytotoxicity and tubular obstruction mediated via bile casts [38, 53, 54]. Bile cast nephropathy or cholemic nephrosis has been largely forgotten in the modern medical literature, a phenomenon which maybe due to the lack of renal biopsy in most pateints with hyperbilirubinemia.Bile casts are analogous to 'myeloma' or myoglobin casts, as they have direct toxic effects on tubular epithelium with an obstructive capacity, which further predispose contrast related tubular injury in CI-AKI.

There are some limitations in this study. First, because this is a retrospective study, we did not regularly collect direct/indirect bilirubin and iron/ferritin (associated with hemoglobin

degradation) before CCT. However, total bilirubin is the most easily obtained data in clinical practice to remind clinicians to avoid CI-AKI. Moreover, it is a novel finding to inform clinicians that hyperbilirubinemia is an independent risk factor for CI-AKI in addition to patients with liver dysfunction. In particular, patients with total bilirubin> 2 mg/dl without baseline liver conditions are easily missed. We highlight this strong association based on current data. If no data of bilirubin, we should also check icteric sclera in our daily practice. More detailed role of direct or indirect bilirubin on renal function after contrast exposure needs further studies in the future. Second, the mechanism of direct injury by severe hyperbilirubinemia in CI-AKI remains to be explored.

## Conclusions

We reported here for the first time, evidence that severe hyperbilirubinemia (total bilirubin > 2.0 mg/dl) is an independent risk factor for CI-AKI, dialysis and mortality after CCT. Most patients with total bilirubin > 2.0mg/dl also had cirrhosis or hepatoma. Clinicians should identify total bilirubin>2.0mg/dl even without cirrhosis or hepatoma.

## Supporting information

**S1 Fig. ROC curve for total bilirubin>1.2 mg/dl to predict acute kidney injury after contrast-enhanced computed tomography.** (AUC = 0.579, with 34.09% of sensitivity and 78.75% of specificity).
(DOC)

**S2 Fig. All patients (n = 9496) and patients with serum bilirubin > 2 mg/dl (n = 1368) divided by different liver conditions.**
(DOC)

**S1 Table. AKI and urgent dialysis within 30 days after contrast-enhanced computed tomography in patients with cirrhosis or hepatoma.**
(DOC)

**S2 Table. Baseline characteristics of patients with total bilirubin> 2mg /dl (n = 1368) divided to liver conditions.**
(DOC)

**S3 Table. Baseline characteristics of patients without cirrhosis or hepatoma (n = 7826) divided to three levels of total bilirubin.**
(DOC)

## Acknowledgments

We thank the Clinical Informatics Research and Development Center of Taichung Veterans General Hospital for assistance in data collection, Miss Lin, Fen-Yi for assistance in data preparation. The authors thank the Biostatistics Task Force of Taichung Veterans General Hospital and Mr. Chen, Jun-Peng for help in statistics.

## Author Contributions

**Data curation:** Yu-Hsien Wu, Chun-Yi Wu, Ching-Yao Cheng.

**Investigation:** Shang-Feng Tsai.

**Resources:** Shang-Feng Tsai.

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
