## [Decision Letter · Decision Letter 0]

23 Jan 2020

PONE-D-19-31096

Severe hyperbilirubinemia is associated with higher risk of contrast-related acute kidney injury following contrast-enhanced computed tomography

PLOS ONE

Dear Prof Shang-Feng Tsai,

Thank you for submitting your manuscript to PLOS ONE. After careful consideration, we feel that it has merit but does not fully meet PLOS ONE’s publication criteria as it currently stands. Therefore, we invite you to submit a revised version of the manuscript that addresses the points raised during the review process.

All reveiwers'comment should be adressed, and the following requests have to be satisfied:

 Hepatoma and cirrhosis are strongly associated with high bilirubin levels and AKI. To show that high bilirubin is an independent risk factor, the authors should also show the incidence of CI-AKI as for figure 2A but only in the 10% patients with high serum total bilirubin levels and no hepatoma or cirrhosis compared to <1.2 and 1.3-2 mg/dl groups

what are the serum levels of conjugated (direct) and unconjugated (indirect) bilirubin? These data can also be informative regarding potential bilirubin conjugation issues leading to CI-AKI.

what are the iron and ferritin levels? Are they associated with CI-AKI?

We would appreciate receiving your revised manuscript by March 31 2020. To enhance the reproducibility of your results, we recommend that if applicable you deposit your laboratory protocols in protocols.io, where a protocol can be assigned its own identifier (DOI) such that it can be cited independently in the future. For instructions see: http://journals.plos.org/plosone/s/submission-guidelines#loc-laboratory-protocols

We look forward to receiving your revised manuscript.

Kind regards,

Olivier Barbier

Academic Editor

PLOS ONE

2. In the ethics statement in the manuscript and in the online submission form, please provide additional information about the patient records/samples used in your retrospective study. Specifically, please ensure that you have discussed whether all data/samples were fully anonymized before you accessed them and/or whether the IRB or ethics committee waived the requirement for informed consent. If patients provided informed written consent to have data/samples from their medical records used in research, please include this information.

'The funders had no role in study design, data collection and analysis, decision to publish, or preparation of the manuscript.'

Please provide an amended Funding Statement that declares *all* the funding or sources of support received during this specific study (whether external or internal to your organization) as detailed online in our guide for authors at http://journals.plos.org/plosone/s/submit-now

Please state what role the funders took in the study.  If any authors received a salary from any of your funders, please state which authors and which funder. If the funders had no role, please state: "The funders had no role in study design, data collection and analysis, decision to publish, or preparation of the manuscript."

c. Please send your amended statements by return email; we will change the online submission form on your behalf.

Reviewers' comments:

Reviewer's Responses to Questions

**Comments to the Author**

1. Is the manuscript technically sound, and do the data support the conclusions?

Reviewer #1: Partly

2. Has the statistical analysis been performed appropriately and rigorously? 

Reviewer #1: Yes

3. Have the authors made all data underlying the findings in their manuscript fully available?

Reviewer #1: Yes

4. Is the manuscript presented in an intelligible fashion and written in standard English?

Reviewer #1: No

5. Review Comments to the Author

Reviewer #1: Wu MJ et al. manuscript is proposing that high serum total bilirubin (> 2mg/dl) could be a risk factor for contrast-related acute kidney injury (CI-AKI). The authors enrolled a total of 9 496 patients who had received contrast-computed tomography and divided them into 3 groups with (normal levels of bilirubin, elevated levels of bilirubin and high bilirubin levels). The authors found that only the patients with high serum total bilirubin (> 2mg/dl) were associated with CI-AKI. Although this group with high bilirubin levels had significantly more cirrhosis or hepatoma contributing to bilirubin levels, the authors suggest that high serum total bilirubin is an independent risk factor for CI-AKI, dialysis and mortality. This is an interesting study with a large number of patients and is differing from other studies suggesting bilirubin is protective for nephropathies. Nevertheless, the current manuscript is only descriptive with associations and no mechanistic insight to understand how bilirubin could contribute to CI-AKI. In addition, only 10% of the patients with high total bilirubin had no cirrhosis or hepatoma which is the most important group to claim that high bilirubin is an independent risk factor. This manuscript was also difficult to follow since the figures are not numbered, figure legends of 3 and 4 are inverted. There are also many syntax and typos errors.

Here are my major concerns:

1. Hepatoma and cirrhosis are strongly associated with high bilirubin levels and AKI. To show that high bilirubin is an independent risk factor, the authors should also show the incidence of CI-AKI as for figure 2A but only in the 10% patients with high serum total bilirubin levels and no hepatoma or cirrhosis compared to <1.2 and 1.3-2 mg/dl groups.

2. This study suggests that patients with high serum total bilirubin levels have a higher risk of CI-AKI of which 16.4% of the patients has developed CI-AKI in this group. Can the authors speculate in the discussion on why a large portion of patients in this group did not develop CI-AKI while others did?

3. Mechanistically, the major source of bilirubin is from heme after hemoglobin degradation. This bilirubin circulates with serum albumin (unconjugated) while the liver conjugates bilirubin to make it soluble for excretion into the bile. The free bilirubin (unconjugated) is the bilirubin that is contributing the most to the effect associated with bilirubin. As such, one missing piece to better understand the protective or deleterious role of bilirubin is the free vs conjugated bilirubin levels. What are the serum levels of conjugated (direct) and unconjugated (indirect) bilirubin? These data can also be informative regarding potential bilirubin conjugation issues leading to CI-AKI.

Along this line, heme is metabolized by heme oxygenase into biliverdin before being converted into bilirubin. These processes release Iron and Carbon monoxide which could reflect also a defect in Heme oxygenase activity. Thus, what are the iron and ferritin levels? Are they associated with CI-AKI?

Minor concerns:

In the discussion, the link between PKC and bilirubin and how it is relevant for the present manuscript is not clear.

6. PLOS authors have the option to publish the peer review history of their article (what does this mean?). If published, this will include your full peer review and any attached files.

Reviewer #1: No

---

## [Author Response · Author response to Decision Letter 0]

19 Feb 2020

Hepatoma and cirrhosis are strongly associated with high bilirubin levels and AKI. To show that high bilirubin is an independent risk factor, the authors should also show the incidence of CI-AKI as for figure 2A but only in the 10% patients with high serum total bilirubin levels and no hepatoma or cirrhosis compared to <1.2 and 1.3-2 mg/dl groups

Thanks for this comment. We re-analyzed this special group (only 10%, without cirrhosis or HCC). We added two more tables in the supplementary file, as table 2 and table 3. 

Supplementary table 2. Baseline characteristics of patients with total bilirubin> 2mg /dl (n=1368) divided to liver conditions. 

Supplementary table 3 Baseline characteristics of patients without cirrhosis or hepatoma (n=7826) divided to three levels of total bilirubin

In supplementary table 3, the incidences of post-contrast AKI in patients without any liver conditions in three different total bilirubin levels were: 8.1% vs. 9.6% vs. 13.4 (p<0.001). We drew this outcome in a new figure 4C to point out the importance of total bilirubin level of all clinical conditions. The result can point out the importance the independent role of total bilirubin >2 on post-contrast AKI.

Thanks for your comments.

What are the serum levels of conjugated (direct) and unconjugated (indirect) bilirubin? These data can also be informative regarding potential bilirubin conjugation issues leading to CI-AKI.

Thanks for this comment. Because this is a retrospective study, we did not regularly collect direct/indirect bilirubin and iron/ferritin (associated with hemoglobin degradation) before contrast-enhanced computed tomography. Indeed, this is a major limitation to our study and we wrote it in the final part of discussion. However, even only with total bilirubin, we can still speculate that there is a merit of this study because of large case numbers and very strong statistical significance after adjusting all comorbidities and laboratory data. In addition, total bilirubin is the most easily obtained data in clinical practice to remind clinicians to avoid contrast related AKI. Moreover, it is a novel finding to inform clinicians that hyperbilirubinemia is an independent risk factor for contrast related AKI in addition to patients with liver dysfunction. This is beyond everyone’s expectation and this point out the importance of this study. We highlight this strong association based on current data. Besides, if no data of bilirubin, we should also check icteric sclera in our daily practice. More detailed role of direct or indirect bilirubin on renal function after contrast exposure will be elucidated in our later study. We added the above explanations in the final part of discussion.

What are the iron and ferritin levels? Are they associated with CI-AKI?

Thanks for your comment. We did not have the data of iron and ferritin because this is a retrospective study. In clinical practice, we did not check iron and ferritin because contrast-enhanced computed tomography. Iron and ferritin levels were associated with hemoglobin degradation, which is suggestive of direct/indirect bilirubin, we will study this part in the later study.

This manuscript was also difficult to follow since the figures are not numbered, figure legends of 3 and 4 are inverted. There are also many syntax and typos errors.

Sorry for this typo. We exchange figure 3 and figure 4. 

1. Hepatoma and cirrhosis are strongly associated with high bilirubin levels and AKI. To show that high bilirubin is an independent risk factor, the authors should also show the incidence of CI-AKI as for figure 2A but only in the 10% patients with high serum total bilirubin levels and no hepatoma or cirrhosis compared to <1.2 and 1.3-2 mg/dl groups. 

Thanks for this comment. We re-analyzed this special group (only 10%, without cirrhosis or HCC). We added two more tables in the supplementary file, as table 2 and table 3. 

Supplementary table 2. Baseline characteristics of patients with total bilirubin> 2mg /dl (n=1368) divided to liver conditions. 

Supplementary table 3 Baseline characteristics of patients without cirrhosis or hepatoma (n=7826) divided to three levels of total bilirubin

In supplementary table 3, the incidences of post-contrast AKI in patients without any liver conditions in three different total bilirubin levels were: 8.1% vs. 9.6% vs. 13.4 (p<0.001). We drew this outcome in a new figure 4C to point out the importance of total bilirubin level of all clinical conditions. The result can point out the importance the independent role of total bilirubin >2 on post-contrast AKI.

Thanks for your comments.

2. This study suggests that patients with high serum total bilirubin levels have a higher risk of CI-AKI of which 16.4% of the patients has developed CI-AKI in this group. Can the authors speculate in the discussion on why a large portion of patients in this group did not develop CI-AKI while others did?

 Because of the incidence of PC-AKI varied according to different baseline conditions. The incidence of PC-AKI in patients with total bilirubin>2.0 mg/dl cannot be considered as too high or too low. The incidence PC-AKI in patients with total bilirubin> 2.0 mg/dl was 16.4%. This is the first study to point out the incidence of PC-AKI regarding hyperbilirubinemia. The incidence of PC-AKI varied from 12 to 50% according to different baseline conditions [2-4, 44-46]. The incidence of PC-AKI in patients with total bilirubin> 2.0 mg/dl maybe relatively low because of the following reasons. First, we recruited both inpatients and outpatients. Overall, this population was all patients receiving CCT with relatively mild disease. Most patients had good baseline renal function (66.6% with baseline eGFR>60 ml/min.1.732m2) and without anemia (12.02 g/dl of mean Hb). Only 30.8% patients had DM and shock was only noticed in 4.0% patients. Second, 33.6% patients had cirrhosis and 26.2% patients had hepatoma. Patients with liver disease may have less skeletal muscle mass, which made less creatine storage, followed by less conversion of creatine to creatinine[47]. Creatinine-based formula may underestimate renal dysfunction[47]. Finally, this is the first study to mention the incidence of PC-AKI in patients with total bilirubin>2.0 mg/dl. Therefore, without previous studies for comparison, the incidence (16.4%) cannot be considered as too high or too low.

We added this discussion in the part of discussion. Thanks for your comment.

3. Mechanistically, the major source of bilirubin is from heme after hemoglobin degradation. This bilirubin circulates with serum albumin (unconjugated) while the liver conjugates bilirubin to make it soluble for excretion into the bile. The free bilirubin (unconjugated) is the bilirubin that is contributing the most to the effect associated with bilirubin. As such, one missing piece to better understand the protective or deleterious role of bilirubin is the free vs conjugated bilirubin levels. What are the serum levels of conjugated (direct) and unconjugated (indirect) bilirubin? These data can also be informative regarding potential bilirubin conjugation issues leading to CI-AKI. Along this line, heme is metabolized by heme oxygenase into biliverdin before being converted into bilirubin. These processes release Iron and Carbon monoxide which could reflect also a defect in Heme oxygenase activity. Thus, what are the iron and ferritin levels? Are they associated with CI-AKI?

Thanks for this comment. Because this is a retrospective study, we did not regularly collect direct/indirect bilirubin and iron/ferritin (associated with hemoglobin degradation) before contrast-enhanced computed tomography. Indeed, this is a major limitation to our study and we wrote it in the final part of discussion. However, even only with total bilirubin, we can still speculate that there is a merit of this study because of large case numbers and very strong statistical significance after adjusting all comorbidities and laboratory data. In addition, total bilirubin is the most easily obtained data in clinical practice to remind clinicians to avoid contrast related AKI. Moreover, it is a novel finding to inform clinicians that hyperbilirubinemia is an independent risk factor for contrast related AKI in addition to patients with liver dysfunction. This is beyond everyone’s expectation and this point out the importance of this study. Besides, if no data of bilirubin, we should also check icteric sclera in our daily practice. We highlight this strong association based on current data. More detailed role of direct or indirect bilirubin on renal function after contrast exposure will be elucidated in our later study. We added the above explanations in the final part of discussion.

Minor concerns:

In the discussion, the link between PKC and bilirubin and how it is relevant for the present manuscript is not clear.

We discussed it because these were studies about the protective function of bilirubin based on protein kinase C (PKC) [Diabetes. 1998;47(6):859-66. American journal of nephrology. 1998;18(4):344-50. Arteriosclerosis, thrombosis, and vascular biology. 1997;17(5):969-78]. However, as your comment, the link was weak and these studies were published many years ago. Thus, we deleted this discussion about the PKC.

---

## [Decision Letter · Decision Letter 1]

12 Mar 2020

PONE-D-19-31096R1

Severe hyperbilirubinemia is associated with higher risk of contrast-related acute kidney injury following contrast-enhanced computed tomography

PLOS ONE

Dear author,

Thank you for submitting your manuscript to PLOS ONE. After careful consideration, we feel that it has merit but does not fully meet PLOS ONE’s publication criteria as it currently stands. Therefore, we invite you to submit a revised version of the manuscript that addresses the points raised during the review process.

Please correct/explain the different in the number of patients as noted by the reviewer.

We would appreciate receiving your revised manuscript by Apr 26 2020 11:59PM. To enhance the reproducibility of your results, we recommend that if applicable you deposit your laboratory protocols in protocols.io, where a protocol can be assigned its own identifier (DOI) such that it can be cited independently in the future. For instructions see: http://journals.plos.org/plosone/s/submission-guidelines#loc-laboratory-protocols

We look forward to receiving your revised manuscript.

Kind regards,

Olivier Barbier

Academic Editor

PLOS ONE

Additional Editor Comments (if provided):

Please correct/explain the different in the number of patients as noted by the reviewer.

Reviewers' comments:

Reviewer's Responses to Questions

**Comments to the Author**

1. If the authors have adequately addressed your comments raised in a previous round of review and you feel that this manuscript is now acceptable for publication, you may indicate that here to bypass the “Comments to the Author” section, enter your conflict of interest statement in the “Confidential to Editor” section, and submit your "Accept" recommendation.

Reviewer #1: (No Response)

2. Is the manuscript technically sound, and do the data support the conclusions?

Reviewer #1: Yes

3. Has the statistical analysis been performed appropriately and rigorously? 

Reviewer #1: Yes

4. Have the authors made all data underlying the findings in their manuscript fully available?

Reviewer #1: Yes

5. Is the manuscript presented in an intelligible fashion and written in standard English?

Reviewer #1: No

6. Review Comments to the Author

Reviewer #1: (No Response)

7. PLOS authors have the option to publish the peer review history of their article (what does this mean?). If published, this will include your full peer review and any attached files.

Reviewer #1: No

---

## [Author Response · Author response to Decision Letter 1]

17 Mar 2020

1.There is a mismatch in the numbers of patients without liver conditions (in the 10% as stated in figure 3). The total number of patients with bilirubin ≥ 2 is n = 1670 page 22 while in the new supplementary table 2 total number of patients with bilirubin ≥ 2 n = 1368. Why is that?

Case numbers with different liver conditions were as follows, 

All patients (n=9496): cirrhosis only (n=659), hepatoma only (n=443), both cirrhosis and hepatoma (n=568), and no cirrhosis no hepatoma (n=7826)

Patients with total bilirubin> 2mg/dl (n=1368): cirrhosis only (n=224), hepatoma only (n=123), both cirrhosis and hepatoma (n=235), and no cirrhosis no hepatoma (n=786)

Therefore, “1670” is cirrhosis only (n=659)+hepatoma only (n=443)+both cirrhosis and hepatoma (n=568) in all patients. We revised the figure legnds of figure 3. Also, we added a supplementary figure 2 (2A for all patients and 2B for patients with total bilirubin>2 mg/dl) . 

2.There are still many typos and syntax errors. 

We revised all text carefully. 

3. This section “However, even only with total bilirubin, we can still speculate that there is a merit of this study because of large case numbers and very strong statistical significance after adjusting all comorbidities and laboratory data.” should be removed from the discussion and the authors should let the readers judge the strength of the manuscript based solely on the data provided. 

According to your suggestion, we deleted it. 

4.All the sections added in the revised manuscript on pages 17-18 should be re-write for a scientific manuscript discussion not as a response to reviewers. 

The discussion should be modified accordingly.

We re-wrote and revised this part. Thanks for your comment. 

Other minors comments:

The authors should decrease the use of “this is the first study”. It is repetitive and unnecessary. 

We decreaed this sentence according to your suggestion. 

Page 18: “Second, the mechanism of direct injury by severe hyperbilirubinemia in CI-AKI remains to be explored. The biological mechanism needs more preclinical studies in the future.” This is redundant…

We deleted “The biological mechanism needs more preclinical studies in the future”

Support of this manuscript: This study was funded by grant TCVGH-1063601B and TCVGH-1073604C from Taichung Veterans General Hospital. The funders had no role in study design, data collection and analysis, decision to publish, or preparation of the manuscript

---

## [Editor Report · Decision Letter 2]

20 Mar 2020

Severe hyperbilirubinemia is associated with higher risk of contrast-related acute kidney injury following contrast-enhanced computed tomography

PONE-D-19-31096R2

Dear Dr. Shang-Feng Tsai,

We are pleased to inform you that your manuscript has been judged scientifically suitable for publication and will be formally accepted for publication once it complies with all outstanding technical requirements.

With kind regards,

Olivier Barbier

Academic Editor

PLOS ONE

---

## [Editor Report · Acceptance letter]

25 Mar 2020

PONE-D-19-31096R2 

Severe hyperbilirubinemia is associated with higher risk of contrast-related acute kidney injury following contrast-enhanced computed tomography 

Dear Dr. Tsai:

I am pleased to inform you that your manuscript has been deemed suitable for publication in PLOS ONE. Congratulations! Your manuscript is now with our production department. 

With kind regards,

on behalf of

Prof. Olivier Barbier 

Academic Editor

PLOS ONE